# Corporate Governance-Based Strategic Approach to Sustainability in Energy Industry of Emerging Economies with a Novel Interval-Valued Intuitionistic Fuzzy Hybrid Decision Making Model

**Wenhao Qi [1], Zhixiong Huang [2,\*], Hasan Dinçer [3], Renata Korsakienė [4] and Serhat Yüksel [3,\*]**

[1]   School of Economics and Management, Jilin Agricultural University, Changchun 130118, China; hao2005buct@jlau.edu.cn

[2]   School of Accounting, ZheJiang University of Finance and Economics, Hangzhou 310018, China

[3]   School of Business, South Campus, İstanbul Medipol University, 34815 Istanbul, Turkey; hdincer@medipol.edu.tr

[4]   Department of Management, Vilnius Gediminas Technical University, Saulėtekio av. 11, LT-10223 Vilnius, Lithuania; renata.korsakiene@vgtu.lt

\*   Correspondence: xboy040552@zufe.edu.cn (Z.H.); serhatyuksel@medipol.edu.tr (S.Y.)

**Abstract:** The sustainability in energy industry is one of the most prominent issues in emerging economies because of needs for the long-term growth of production and managerial capacity. Accordingly, corporate governance could lead to develop the sustainable production of energy industry. The purpose of this study is to define a set of criteria and dimensions for analyzing the corporate governance-based strategic approach to sustainability in the energy industry of emerging economies. For this purpose, this study provides several novelties by extending a hybrid decision making model with interval-valued intuitionistic fuzzy sets (IVIF) and defining the related criteria and dimensions of corporate governance-based strategic approach with the supported literature. IVIF decision making trial and evaluation laboratory (DEMATEL) is constructed for measuring the relative importance of criteria and dimensions. IVIF VlseKriterijumska Optimizacija I Kompromisno Resenje (VIKOR) is applied for ranking the corporate governance-based performance of sustainable energy industries in emerging economies. Sensitivity analysis is also used for understanding the coherence of ranking results. Analysis results illustrate that the energy industry could provide more sustainable results than the conventional managerial policies by considering the social capital of board members. Additionally, mass-economies are closely related to the sustainable production capacities of energy industry and have the best performance results for the corporate governance-based sustainable energy production strategies. The results are discussed to provide the policy recommendations by comparing analysis results of emerging economies for further studies.

**Keywords:** sustainability; energy industry; production; emerging economies; corporate governance; IVIF DEMATEL; IVIF VIKOR

## 1. Introduction

In an emerging countries context, energy industry significantly contributes to economic growth [1]. On the other hand, it is important to meet current requirements and consider future demands of upcoming generations. Thus, the companies operating in energy industry are faced with a number of challenges. First, energy consumption in emerging countries, such as Brazil, China, India, South Africa

is higher as compared to developed countries due to growing population. Second, the industry is using traditional energy sources which contribute to the environmental pollution. Subsequently, the policies and legislation of these countries have been changing the development directions of the energy industry by promoting sustainable energy production.

The concerns about corporate sustainability performance switched the predominating emphasis on profit to social, environmental, voluntariness, and stakeholder dimensions of business [2] and, thus, the companies were forced to set new sustainable objectives and align new development directions with the core business. Moreover, the motivation of the companies to integrate sustainability issues into their strategies rather than following strict regulations is observed [3]. Within this context, the term "green governance", which emphasizes the actions, supporting green environment, and application of governance mechanisms, impacting green practices, has emerged [4]. Thus, it is recognized as the catalyst for corporate governance, responsible for protection of stakeholders' interests and support in decision-making processes [5]. The impact of a board of directors on corporate sustainability has been recognized as an emerging research question. The studies demonstrate that more independent boards better represent stakeholders' interests [6] and, thus, the sustainability is observed [7].

While sustainability refers to economic, social, and environmental sustainability [8], sustainable production is assumed to be production that is socially beneficial [9]. The energy companies must deal with high complexity due to conflicting goals and interests such as profit, market competitiveness, social, and environmental impact. The corporate governance and in particular boards of directors appear to be a mechanism through which the companies deal with such complexity. Thus, the changes of board members and the focus on individuals who are capable of contributing to proactive sustainability strategy and monitoring the implementation of sustainable production philosophy in the business environment have become the necessity. The understanding of board specific characteristics, impacting the appointment of new members in energy industry, appears to be a promising research area.

Efficient use of energy and sustainable production of energy industry are among the most important factors for emerging economies. Understanding managerial capacities and cooperation with governance could boost the production potential of energy industry and the determinants of corporate governance should be integrated for measuring the sustainable production of energy industry. Accordingly, energy industry of emerging economies is analyzed by using the criteria of corporate governance-based strategic priority for the sustainable production. Therefore, in this study we aim to define the issues for analyzing the corporate governance-based strategic approach to sustainability in the energy industry of emerging economies. Accordingly, the study addresses the following research questions: (1) what are specific characteristics of new board members contributing to sustainable production in energy industry? (2) How can specific characteristics of new board members contributing to sustainable production in energy industry be assessed? (3) What are the rankings of E7 countries considering board specific characteristics contributing to sustainable energy production? The study is novel and different from the approaches applied in the scientific studies. While a number of studies focus on some aspects of board composition such as independence, diversity, etc. [10], our study considers a more comprehensive range of characteristics. Moreover, the study applies multiple criteria decision analysis (MCDA) methods, which identify criteria and integrate these criteria in the decision-making process. These methods have become popular in sustainability research [11] and high-skilled human resources field [12,13]. Thus, in answering the research questions, we apply IVIF DEMATEL for weighting the criteria and dimensions and IVIF VIKOR for ranking the emerging countries. The main benefit of considering IVIF sets is minimizing uncertainties in the decision-making process. On the other hand, the relationship between the criteria can be evaluated owing to the DEMATEL methodology. Additionally, the advantage of the VIKOR method is that more appropriate results can be reached because of considering the utility and regret.

In the first part, the theoretical discussion on sustainability, sustainable production in energy industry, and interrelationships of corporate governance and sustainable production is provided. The next section provides the methodological background. IVIF sets, IVIF DEMATEL, and IVIF VIKOR

are presented. The fourth section presents the application of a hybrid decision making model and obtained results. The final section provides discussion and conclusions.

## 2. Theoretical Background

### 2.1. Sustainability and Sustainable Production in Energy Industry

Energy is very important in terms of social and economic development in a country. Thanks to energy, people can meet their daily needs such as warming. Therefore, it is accepted that there is a strong connection between energy and people's social happiness. In addition to the mentioned issue, energy plays an important role in the economic development of countries [14]. The main reason for this is that energy is a very important raw material of the industry. As can be seen here, there is a positive correlation between a country's energy consumption and the amount of production. Therefore, more energy is needed to increase the industrial production of countries. In this way, new investments will increase, and this will contribute to the economic growth of the country [15]. Another advantage of the mentioned issue is that it is possible to reduce the unemployment rate of the country by providing new employment opportunities. It is obvious that the energy industry has a significant impact on sustainable development [16].

In this process, one of the most used energy types is non-renewable energies. In this context, these types of energy are formed as a result of burning fossil fuels. Oil, natural gas, and nuclear are important examples of non-renewable energy sources. One of the most important reasons for preferring these types of energy is low cost. On the other hand, easier storage of reserves is also an important source of choice for non-renewable energies. However, these types of energy also have a number of disadvantages. The extensive use of non-renewable natural resources significantly influences high emission levels, especially in industrialized countries and subsequently contribute to the warming climate [17]. Meanwhile, developing countries are facing increased levels of emission due to economic development and population growth. Sustainability in energy industry refers to energy sources which have minimal environmental impact while considering social dimension through reliable and affordable energy supplies [18].

In addition, renewable energy sources take their sources from nature like wind and sun. This situation has many advantages to the country. For example, since renewable energy sources do not release carbon into the atmosphere, they do not create environmental pollution. Thus, the sustainable energy production appears to be a solution decreasing the negative impact on human health and contributing to the vital ecological systems. The focus on sustainable energy production has switched the attention of both scholars and practitioners to renewable energy due to a great amount of potential as well as economic considerations. Sustainable energy production is seen as a way to respond to previous conflicts [19].

On the firm level, energy companies are pressured to respond to sustainability issues such as environmental performance, social impact, resource efficiency, and others [20]. It appears that the firms are facing a challenge to implement short-term and long-term objectives. First, the energy companies are expected to follow many guidelines, international initiatives, and agreements to gain trust of key stakeholders [18]. Second, the energy companies are expected to remain profitable, competitive, and maintain the social license to operate [21]. Moreover, scholars observe that challenges for sustainable production in energy industry are linked to insufficient human resources, failure to involve the beneficiaries, and excessive focus on tactical solutions. Given this context, the role of those entrusted with corporate governance appear to be significant in balancing conflicting goals. The scholars argue that the board of directors make a number of sustainability related decisions [22], however, their role in energy industry requires deeper understanding.

*2.2. Corporate Governance and Sustainable Production*

The studies on boards of directors originated from the wider corporate governance field and attracted a number of scholars in the last few decades. The literature assumes that boards of directors are one of the most significant governance mechanisms [23] and plays a significant role irrespective of the firm's size and business sector. For instance, the investigations performed in the energy industry demonstrate the assignment of sustainability initiatives to the supervisory boards [24]. The independent directors involved in the corporate governance may assure higher performance. Thus, the studies confirmed that outside and independent directors positively influenced value creation in terms of financial and non-financial outcomes [25]. The constant pressures to consider sustainability issues encourage energy companies to adopt strategies leading to enhanced environmental performance. The studies confirmed the direct and significant effect of boards on carbon performance of the firms through carbon strategy in high carbon-intensive industries, such as oil and gas [20], what led to the conclusion that greenhouse gas emissions became an integral part of the companies' sustainability strategy. The literature adopted a multi-theoretical framework embracing different roles of boards of directors and, thus, the discussion on these theories and possible interrelationships with sustainable production in energy companies are provided below.

The integration of sustainable production of energy companies into the overall organizational strategy is usually related to the costs due to adoption of new technologies and processes. Apparently, long-term perspective is needed for the benefits of enhanced environmental performance [26]. Thus, stemming from agency theory, the control role of board members refers to monitoring of management decisions and protection of shareholders' interests [27]. The effectiveness of control may be achieved through enquiries about strategic directions adopted by the top management. The investigations reveal that companies with active and independent boards tend to consider environmental issues [23]. Thus, we raise the assumption that the board members and their characteristics are the major factor impacting sustainable production of energy companies in spite of significant costs. Previous studies demonstrate the importance of board members for sustainable production. For instance, the studies highlighted that independent directors with multiple directorships experience may significantly influence environmental policy and practice [28].

Resource dependence theory suggests provision of legitimization, expert advice, counselling, or links to other organizations [29]. The directors are assumed to be a resource which can be classified considering their roles such as, "support specialist", "business experts", and "community influences". Moreover, the strategic role of boards emphasizes a strategic contribution of boards to the firm through formulation, refinement, and evaluation strategies [30]. The board of directors may provide informal advice and guide strategic changes which lead to sustainable production of energy companies. Social capital of board members is embedded in social ties and stems from the networks of relationships [31]. Thus, the social background of board members appears to be significant for the monitoring, counselling, and advice role. On the other hand, board members should have cognitive abilities which lead to the recognition of opportunities and threats [10]. The studies suggest that multiple directorships of board members lead to the understanding of governance and strategic issues of other companies related to environmental practices and subsequently to sustainable production. For instance, the investigation of US electric power firms disclosed that industry diversity ties of boards were positively related to environmental performance [32].

Grounded in legitimacy theory, the studies emphasize social and environmental responsibilities of the firms contributing to the improved legitimacy. Considering the aim of the energy companies to maintain their legitimacy through enhanced environmental performance, we assume that sustainable production may lead to enhanced reputation and elimination of negative consequences such as boycotts. Some investigations have paid attention to the mediation role of boards between the firms and their stakeholders [33]. The diversity of strategic interests appears to be a key determinant impacting market positioning of energy companies [22]. The stakeholder's theory suggests that the board members assure the systematic balance between the interests of shareholders and stakeholders. Recent studies

have emphasized a significant role of the board members in responding to the sustainability issues of the firms [5] and sustainability performance [7]. Apparently, formulation of effective strategy requires diverse expertise [34] and a wide understanding of stakeholders' needs. For instance, a number of studies revealed a positive relationship of education and environmental aspects [7]. The explanation resides in the fact that advanced education of board members leads to the better perceptions of environmental issues. Thus, the advanced education leads to the acquisition of "green" skills and competencies which help to deal with various societal issues [35].

Table 1 summarizes the similar studies in the literature. For the human capital dimension, advanced education plays a key role in improving corporate governance [7]. Additionally, environmental experience can also contribute to this situation [35]. Another important criterion in this framework is the multiple directorships experience [8]. On the other side, regarding the social capital dimension, strong ties to the political organizations and universities can provide effective corporate governance [36–38]. Similar to this situation, the companies, which aim to improve corporate governance, should have effective ties to other companies [32]. Moreover, sensing and seizing abilities are also significant in this framework [39,40]. Finally, reconfiguring abilities in the company can be very helpful for improving the effectiveness in the corporate governance activities [31].

**Table 1.** Proposed dimensions and criteria of corporate governance for sustainable production.

| Dimension | Criterion | Supporting Literature |
|---|---|---|
| Human capital (Dimension 1) | Advanced education (Criterion 1) | [7,36,37] |
| | Environmental experience (Criterion 2) | [35] |
| | Multiple directorships experience (Criterion 3) | [8] |
| Social capital (Dimension 2) | Ties to political organizations (Criterion 4) | [38] |
| | Ties to universities (Criterion 5) | [38] |
| | Ties to other companies (Criterion 6) | [32] |
| Cognitive capabilities (Dimension 3) | Sensing abilities (Criterion 7) | [31,39] |
| | Seizing abilities (Criterion 8) | [40] |
| | Reconfiguring abilities (Criterion 9) | [33] |

### 2.3. Literature on MCDM Methods

A considerable number of studies investigated a limited number of board characteristics, what lets us conclude that more integrated approach in evaluation of new board members to transition toward the sustainable production in energy industry is needed. MCDA methods appear to be popular in solving complex issues in the high-skilled human resources field. However, the analysis of the literature revealed no prior application of MCDA methods in such specific area as presented in this study. For instance, Krishankumar et al. [13] focused on the personnel selection problem for information technologies industry. In the analysis process, extended VIKOR under intuitionistic fuzzy set context and intuitionistic fuzzy AHP were considered. Karabasevic et al. [39] also undertook a similar examination for the same industry with EDAS and SWARA.

Sang et al. [40] undertook an analysis for the personnel selection problem. In this framework, knowledge-intensive enterprises were included in the scope of the analysis. The evaluation of these companies was carried out by Karnik–Mendel (KM) algorithm and fuzzy TOPSIS approaches. Parallel to this study, Kelemenis et al. [41] also tried to undertake an evaluation regarding effective personnel selection. On the other side, Kabak [42] aimed to identify the significant points in the selection of personnel. This analysis has been performed for militaries. In this context, a fuzzy DEMATEL-ANP based method is used for the evaluation process of this study. It is found that there is a strong relationship between the corporate governance and sustainable production. Hence, it is believed that sector-based evaluation can contribute the literature in this framework. For this purpose, an evaluation was performed in this study for energy industry. Another important point is that there are limited studies in which MCDM methods are considered for the energy industry. Thus,

by considering IVIF DEMATEL and IVIF VIKOR methods for energy industry in this study, we aimed to improve originality.

## 3. Methodology

### 3.1. IVIF Sets

By the 1960s, the concept of fuzzy logic was introduced by Zadeh [43] for understanding the complicated topic of the real-world problems. Especially, Atanassov extended the methodology using the intuitionistic sets and defined these as the IVIF sets [44]. It considers the membership and non-membership degrees for the items of relation and decision matrices accordingly [45]. Thus, it is possible to find out the more accurate results under the fuzzy environment. The general information of IVIF sets are summarized as follows.

$$I = \{\langle \vartheta, \mu_I(\vartheta), n_I(\vartheta) \rangle / \vartheta \epsilon U\} \tag{1}$$

$I$ is Intuitionistic fuzzy set, $\mu_I(\vartheta) : U \rightarrow [0,1]$, $n_I(\vartheta) : U \rightarrow [0,1]$, and $0 \leq \mu_I(\vartheta) + n_I(\vartheta) \leq 1$.
$\mu_I(\vartheta)$ belongingness and non-belongingness $n_I(\vartheta)$ are the degrees of $\vartheta$. $\mu_{IU}(\vartheta), n_{IU}(\vartheta), \mu_{IL}(\vartheta),$ $n_{IL}(\vartheta)$ are defined as the upper and lower values of $\mu_I(\vartheta)$ and $n_I(\vartheta)$, respectively.

$$I = \{\vartheta, [\mu_{IL}(\vartheta), \mu_{IU}(\vartheta)], [n_{IL}(\vartheta), n_{IU}(\vartheta)] / \vartheta \epsilon U\} \tag{2}$$

where

$$\begin{aligned} 0 \leq \mu_{IU}(\vartheta) + n_{IU}(\vartheta) &\leq 1 \\ \mu_{IL}(\vartheta) &\geq 0 \\ n_{IL}(\vartheta) &\geq 0 \end{aligned} \tag{3}$$

$$\tau_I(\vartheta) = 1 - \mu_I(\vartheta) - n_I(\vartheta) \tag{4}$$

where $\tau_I(\vartheta)$ is the unknown degree of IVIF sets.

However, the belongingness and non-belongingness degrees of IVIF set are illustrated as in Equation (5). In this equation, $a$ and $b$ represent the upper and lower values of the belongingness whereas these values for non-belongingness are named as $c$ and $d$.

$$I = ([a, b], [c, d]) \tag{5}$$

### 3.2. IVIF DEMATEL

DEMATEL measures the direct relation degrees between factors. The DEMATEL gives more prominent results to illustrate the influence degrees and directions between each other, even though there are several approaches to employ the weights of factors [45]. In recent literature, there are several extensions of DEMATEL to increase the accuracy of computation, the use of IVIF sets is a novel approach to the DEMATEL extensions. Accordingly, the formulization of the IVIF DEMATEL is given in the following steps.

The first step is to obtain the linguistic choices for each relationship between the criteria and dimensions. For that, the linguistic evaluations are defined in the form of IVIF numbers by Equation (6)

$$\widetilde{Z}_{ij} = \left( \left( a_{ij}, b_{ij} \right), \left( c_{ij}, d_{ij} \right) \right) \tag{6}$$

$a_{ij}$ and $c_{ij}$ are lower, $b_{ij}$ and $d_{ij}$ are the upper values of belongingness and non-belongingness degrees, respectively. The matrix is presented in Equation (7)

$$
\widetilde{Z} = \begin{bmatrix}
0 & \widetilde{z}_{12} & \cdots & \cdots & \widetilde{z}_{1n} \\
\widetilde{z}_{21} & 0 & \cdots & & \cdots & \widetilde{z}_{2n} \\
\vdots & \vdots & \ddots & \cdots & \cdots \\
\vdots & \vdots & \vdots & \ddots & \vdots \\
\widetilde{z}_{n1} & \widetilde{z}_{n2} & \cdots & \cdots & 0
\end{bmatrix}
\tag{7}
$$

The averaged scores of expert evaluations are employed to construct the direct relation matrix as seen in Equation (8)

$$
\widetilde{Z} = \frac{\widetilde{Z}^1 + \widetilde{Z}^2 + \widetilde{Z}^3 + \cdots \widetilde{Z}^n}{n}
\tag{8}
$$

The second step is the normalization process of the matrix with Equations (9)–(11)

$$
\widetilde{X} = \begin{bmatrix}
\widetilde{x}_{11} & \widetilde{x}_{12} & \cdots & \cdots & \widetilde{x}_{1n} \\
\widetilde{x}_{21} & \widetilde{x}_{22} & \cdots & & \cdots & \widetilde{x}_{2n} \\
\vdots & \vdots & \ddots & \cdots & \cdots \\
\vdots & \vdots & \vdots & \ddots & \vdots \\
\widetilde{x}_{n1} & \widetilde{x}_{n2} & \cdots & \cdots & \widetilde{x}_{nn}
\end{bmatrix}
\tag{9}
$$

where

$$
\widetilde{x}_{ij} = \frac{\widetilde{z}_{ij}}{r} = \left[ \left( \frac{Z_{a'_{ij}}}{r}, \frac{Z_{b'_{ij}}}{r} \right), \left( \frac{Z_{c'_{ij}}}{r}, \frac{Z_{d'_{ij}}}{r} \right) \right]
\tag{10}
$$

$$
r = max\left( max_{1 \leq i \leq n} \sum_{i=1}^{n} Z_{b'_{ij}}, max_{1 \leq j \leq n} \sum_{j=1}^{n} Z_{b'_{ij}} \right)
\tag{11}
$$

The third step is to compute the total relation matrix in the form of IVIF sets by Equations (12)–(16)

$$
X_{a'} = \begin{bmatrix}
0 & a'_{12} & \cdots & \cdots & a'_{1n} \\
a'_{21} & 0 & \cdots & & \cdots & a'_{2n} \\
\vdots & \vdots & \ddots & \cdots & \cdots \\
\vdots & \vdots & \vdots & \ddots & \vdots \\
a'_{n1} & a'_{n2} & \cdots & \cdots & 0
\end{bmatrix}
$$

$$
X_{d'} = \begin{bmatrix}
0 & d'_{12} & \cdots & \cdots & d'_{1n} \\
d'_{21} & 0 & \cdots & & \cdots & d'_{2n} \\
\vdots & \vdots & \ddots & \cdots & \cdots \\
\vdots & \vdots & \vdots & \ddots & \vdots \\
d'_{n1} & d'_{n2} & \cdots & \cdots & 0
\end{bmatrix}
\tag{12}
$$

$$
\widetilde{T} = \lim_{k \to \infty} \widetilde{X} + \widetilde{X}^2 + \cdots + \widetilde{X}^k
\tag{13}
$$

$$
\widetilde{T} = \begin{bmatrix}
\widetilde{t}_{11} & \widetilde{t}_{12} & \cdots & \cdots & \widetilde{t}_{1n} \\
\widetilde{t}_{21} & \widetilde{t}_{22} & \cdots & & \cdots & \widetilde{t}_{2n} \\
\vdots & \vdots & \ddots & \cdots & \cdots \\
\vdots & \vdots & \vdots & \ddots & \vdots \\
\widetilde{t}_{n1} & \widetilde{t}_{n2} & \cdots & \cdots & \widetilde{t}_{nn}
\end{bmatrix}
\tag{14}
$$

where

$$\widetilde{t}_{ij} = \left( \left( a''_{ij}, b''_{ij} \right), \left( c''_{ij}, d''_{ij} \right) \right) \tag{15}$$

$$\left[ a''_{ij} \right] = X_{a'} \times (I - X_{a'})^{-1}, \ \left[ d''_{ij} \right] = X_{d'} \times (I - X_{d'})^{-1} \tag{16}$$

The following step is to compute the values of $\widetilde{D}$ and $\widetilde{R}_i$. The values are determined by summing the vector rows and columns of the total relation matrix with Equations (17) and (18)

$$\widetilde{D}_i = \left[ \sum_{j=1}^{n} \widetilde{t}_{ij} \right]_{n \times 1} \tag{17}$$

$$\widetilde{R}_i = \left[ \sum_{i=1}^{n} \widetilde{t}_{ij} \right]'_{1 \times n} \tag{18}$$

however, $\left( \widetilde{D}_i + \widetilde{R}_i \right)$ is the weights of the factors while $\left( \widetilde{D}_i - \widetilde{R}_i \right)$ is the influencing directions and degrees among the factors. Accuracy function $H(i)$ is applied to obtain the weights of criteria as seen in Equation (19)

$$H(i) = \frac{a + b + c + d}{2} \quad H(i) \in [0, 1] \ [a, b][c, d] \tag{19}$$

These are the items of IVIF sets.

### 3.3. IVIF VIKOR

VIKOR considers the best and worst values of fuzzy numbers with the maximum group utility. In addition, overall results are checked to understand whether the acceptable advantages and stability are provided. Thus, the VIKOR method is frequently preferred to rank the alternatives under complex decision making problems with several extensions [46]. The IVIF VIKOR method is proposed to measure the ranking performance of alternatives more effectively by considering the advantages of IVIF sets. This extension is summarized by the following steps. Firstly, the fuzzy decision matrix is illustrated by Equations (20) and (21).

$$D = \begin{matrix} A_1 \\ A_2 \\ A_3 \\ \vdots \\ A_m \end{matrix} \begin{bmatrix} h_{11} & h_{12} & h_{13} & \cdots & h_{1n} \\ h_{21} & h_{22} & h_{23} & \cdots & h_{2n} \\ h_{31} & h_{32} & h_{33} & \cdots & h_{3n} \\ \vdots & \vdots & \vdots & \ddots & \vdots \\ h_{m1} & h_{m2} & h_{m3} & \cdots & h_{mn} \end{bmatrix} \tag{20}$$

$A_m$ defines the alternatives whereas $C_n$ is the criterion set. Meanwhile, $h_{ij}$ gives the decision evaluations with the IVIF numbers for each alternative. However, the final decision matrix is constructed with the averaged values of the expert scores as seen in Equation (21).

$$h_{ij} = \frac{1}{k} \left[ \sum_{e=1}^{n} h_{ij}^{e} \right], \ i = 1, m; \ j = 1, n \tag{21}$$

The next step is to calculate the best ($f_j^*$) and worst ($f_J^-$) values of the decision matrix as

$$f_J^* = \overset{max}{\underset{i}{}} x_{ij} \ and \ f_J^- = \overset{min}{\underset{i}{}} x_{ij} \tag{22}$$

The third step is to compute $S_i$, $R_i$, and $Q_i$ values by Equations (23)–(25)

$$S_i = \sum_{i=1}^{n} w_j \frac{\left(\left|f_j^* - x_{ij}\right|\right)}{\left(\left|f_j^* - f_j^-\right|\right)} \tag{23}$$

$$R_i = \max_j \left[ w_j \frac{\left(\left|f_j^* - x_{ij}\right|\right)}{\left(\left|f_j^* - f_j^-\right|\right)} \right] \tag{24}$$

$$Q_i = \frac{v(S_i - S^*)}{(S^- - S^*)} + (1 - v)\frac{(R_i - R^*)}{(R^- - R^*)} \tag{25}$$

In this process, $v$ indicates the maximum group utility. Furthermore, two different conditions should be achieved as in Equations (26) and (27).

$$Q\left(A^{(2)}\right) - Q\left(A^{(1)}\right) \geq 1/(j-1) \tag{26}$$

$$Q\left(A^{(M)}\right) - Q\left(A^{(1)}\right) < \frac{1}{(j-1)} \tag{27}$$

$A^{(2)}$ is alternative that has the second ranking result and $j$ is the alternative numbers. Additionally, $A^{(m)}$ explains the number of alternatives. The final step is to rank the alternatives in ascending order according to the Qi results. Similarly, the defuzzification procedure is applied by considering the accuracy function of the decision matrix.

## 4. Case Study

IVIF DEMATEL is used for weighting the factors of the corporate governance and IVIF VIKOR is applied for evaluating the emerging economy alternatives. The main advantage of using IVIF sets is that it can be possible to minimize uncertainty in the analysis process Additionally, owing to the DEMATEL approach, an impact relation map can be generated. Moreover, compromise solutions can be achieved by considering the closeness to the ideal solutions.

This analysis is constructed in the following steps and the results are summarized, respectively, by using the proposed IVIF-based hybrid decision making model. In the first step, a set of criteria and dimensions of the board of directors promoting sustainable energy production in energy industry is defined. Table 1 illustrates these items.

Step 2: Collect the expert evaluations for the factors and alternatives. Thus, the expert team including three decision makers is selected aiming to obtain evaluations of the factors and alternatives. Selected decision makers are managers and work in the field of global energy industry as well as emerging economies with at least 10 years of industry experience. They provide their evaluations with linguistic scores. Table 2 shows the linguistic scores and evaluation percentages for analyzing the criteria, dimensions, and alternatives.

Table 2 indicates the scores for linguistic evaluations. In the IVIF sets, belongingness and non-belongingness are taken into account in the analysis process. The total of them should be equal to 1. For this purpose, in Table 2, percentages are proposed for different linguistic terms for this framework. Decision makers give their linguistic priorities for each relation between criteria and dimensions. Tables 3–6 represent the evaluations for the dimensions and their criteria. Table 3 mainly gives information about the linguistic evaluations for the dimensions. For this purpose, three different experts made comparative evaluations to understand which dimensions are more significant. Similarly, Tables 4–6 explain these examinations for criteria under each of the three dimensions.

**Table 2.** Scores for linguistic evaluations.

| Linguistic Terms | Percentages |
|:---:|:---:|
| Very Low (VL) | 0.1 |
| Low (L) | 0.2 |
| Medium Low (ML) | 0.3 |
| Medium (M) | 0.4 |
| Medium High (MH) | 0.5 |
| High (H) | 0.6 |
| Very High (VH) | 0.7 |
| Absolute (A) | 0.8 |

In addition, the decision makers provide their evaluations for constructing the decision matrix. Table 7 shows the linguistic results of alternatives with respect to criteria.

Step 3: Weight the dimensions and criteria. The IVIF DEMATEL method is applied for weighting the criteria of the board of directors promoting sustainable energy production in energy industry. For this purpose, the computation process of IVIF DEMATEL is considered, respectively, by using Equations (6)–(19). Tables 8–10 illustrate the results of extended method for the criteria and dimensions.

Similar procedures are also applied for the criteria of each dimension and then the weights of criteria and dimensions are calculated. The results are given in Table 11. In this table, both local and global weights are identified. The local weights explain the significance levels of the criteria in their own dimension. On the other side, global weights make a cumulative analysis and state these importance levels while considering whole criteria.

According to the results, social capital (dimension 2) has the highest importance among the dimensions, followed by cognitive capabilities (dimension 3) and human capital (dimension 1). However, the global weights of criteria illustrate that ties to other companies (Criterion 6) has the highest importance among criteria followed by reconfiguring capabilities (Criterion 9) and ties to universities (Criterion 6). In addition to them, multiple directorships (Criterion 3) and ties to political organizations (Criterion 4) are other significant criteria among the top five characteristics of the board of directors promoting sustainable energy production in energy industry. On the other side, it is also determined that advanced education (Criterion 1), sensing abilities (Criterion 7), and seizing abilities (Criterion 8) play a lower role in comparison with the other ones.

The extended VIKOR method based on IVIFs is considered to measure the ranking results of energy industries of emerging economies. The competition procedures given by Equations (20)–(27) are used for ranking the emerging economy alternatives. Tables 12–14 present the computation results of the VIKOR method accordingly.

The values of $Q_i$ are listed in ascending order to measure the rankings of energy industries of E7 countries considering board specific characteristics promoting sustainable energy production. Accordingly, the ranking results demonstrate that alternative 1 (China) has the best rank among the economies. Additionally, A7 (Turkey) and A4 (Mexico) are other successful countries for this purpose. Nevertheless, it is also concluded that alternative 6 (Indonesia) takes the worst place among the emerging economies. Similarly, A5 (Russia) and A3 (Brazil) are also less successful by compared with others. However, sensitivity analysis is also applied for illustrating the coherencies of importance degrees for the criteria. Table 15 represents the sensitivity analysis for nine cases.



**Table 3.** Input data for the dimensions.

| | D1 | | | D2 | | | D3 | | |
| --- | --- | --- | --- | --- | --- | --- | --- | --- | --- |
| | DM1 | DM2 | DM3 | DM1 | DM2 | DM3 | DM1 | DM2 | DM3 |
| D1 | - - - - | - - - - | - - - - | M H VL ML | ML MH L M | ML MH L M | L MH L ML | VL MH ML M | VL ML VL L |
| D2 | M VH L ML | ML ML MH L | M M MH ML M | - - - - | - - - - | - - - - | M VH L ML | ML ML MH L | M M MH ML M |
| D3 | M H VL ML M | MH ML M | H VH VL L | M VH L ML | ML ML MH L | M M MH ML M | - - - - | - - - - | - - - - |

**Table 4.** Input data for the criteria of dimension 1.

| | C1 | | | C2 | | | C3 | | |
| --- | --- | --- | --- | --- | --- | --- | --- | --- | --- |
| | DM1 | DM2 | DM3 | DM1 | DM2 | DM3 | DM1 | DM2 | DM3 |
| C1 | - - - - | - - - - | - - - - | M MH ML M | ML MH VL M | M MH ML M | M H VL ML | M MH ML M | H VH VL L |
| C2 | M H VL ML | VL ML VL L | ML MH L M | - - - - | - - - - | - - - - | M VH L ML | ML ML MH L | M M MH ML M |
| C3 | L MH L ML | VL MH ML M | VL ML VL L | M H VL ML | ML ML MH L | M M MH ML M | - - - - | - - - - | - - - - |

**Table 5.** Input data for the criteria of dimension 2.

| | C4 | | | C5 | | | C6 | | |
| --- | --- | --- | --- | --- | --- | --- | --- | --- | --- |
| | DM1 | DM2 | DM3 | DM1 | DM2 | DM3 | DM1 | DM2 | DM3 |
| C4 | - - - - | - - - - | - - - - | M H VL ML | ML MH VL M | M H VL ML | H VH VL L | M VH L ML | H VH VL L |
| C5 | VL MH ML M | VL ML VL L | VL MH ML M | - - - - | - - - - | - - - - | M VH L ML | H VH VL L | M MH ML M |
| C6 | VL MH ML M | VL MH ML M | VL ML VL L | ML MH L M | ML MH L M | ML MH L M | - - - - | - - - - | - - - - |

**Table 6.** Input data for the criteria of dimension 3.

| | C7 | | | C8 | | | C9 | | |
| --- | --- | --- | --- | --- | --- | --- | --- | --- | --- |
| | DM1 | DM2 | DM3 | DM1 | DM2 | DM3 | DM1 | DM2 | DM3 |
| C7 | - - - - | - - - - | - - - - | M H VL ML | ML MH L M | ML MH L M | M MH ML M | M MH ML M | M VH L ML |
| C8 | M H VL ML | VL ML VL L | ML MH L M | - - - - | - - - - | - - - - | M VH L ML | ML ML MH L | M M MH ML M |
| C9 | M H VL ML | VL MH ML M | M H VL ML | M H VL ML | ML ML MH L | M M H VL ML | - - - - | - - - - | - - - - |

**Table 7.** Input data for the alternatives.

| | Alternative 1 (China) | | | Alternative 2 (India) | | | Alternative 3 (Brazil) | | |
|---|---|---|---|---|---|---|---|---|---|
| | DM1 | DM2 | DM3 | DM1 | DM2 | DM3 | DM1 | DM2 | DM3 |
| C1 | MH VH VL L | MH VH VL L | ML VH L ML | ML VH L ML | ML VH L ML | MH VH L ML | MH VH VL L | VL M L ML | H A VL L |
| C2 | VL M L ML | VL M L ML | H A VL L | ML VH L ML | ML MH ML M | VL M L ML | VL M L ML | H A VL L | ML MH ML M |
| C3 | H A VL L | H A VL L | ML H ML M | MH VH VL L | VL M L ML | H A VL L | H A VL L | ML MH ML M | ML H ML M |
| C4 | ML MH ML M | ML H ML M | MH VH VL L | H A VL L | H A VL L | ML MH ML M | H A VL L | VL M L ML | ML VH L ML |
| C5 | ML H ML M | ML VH L ML | MH VH VL L | ML VH L ML | H A VL L | VL M L ML | ML VH L ML | ML MH ML M | VL M L ML |
| C6 | ML VH L ML | H A VL L | ML VH L ML | H A VL L | ML MH ML M | ML MH ML M | ML MH ML M | ML MH ML M | H A VL L |
| C7 | H A VL L | ML H ML M | ML MH ML M | ML MH ML M | ML MH ML M | VL M L ML | H A VL L | VL M L ML | ML MH ML M |
| C8 | ML H ML M | MH VH VL L | H A VL L | VL M L ML | ML MH ML M | H A VL L | H A VL L | H A VL L | VL M L ML |
| C9 | MH VH VL L | ML H ML M | ML H ML M | ML MH ML M | H A VL L | H A VL L | H A VL L | MH VH VL L | ML MH ML M |

| | Alternative 4 (Mexico) | | | Alternative 5 (Russia) | | | Alternative 6 (Indonesia) | | |
|---|---|---|---|---|---|---|---|---|---|
| | DM1 | DM2 | DM3 | DM1 | DM2 | DM3 | DM1 | DM2 | DM3 |
| C1 | MH VH VL L | VL M L ML | ML VH L ML | ML VH L ML | ML VH L ML | MH VH VL L | H A VL L | VL M L ML | ML VH L ML |
| C2 | VL M L ML | ML MH ML M | H A VL L | ML MH ML M | VL M L ML | VL M L ML | ML MH ML M | ML MH ML M | ML H ML M |
| C3 | H A VL L | VL M L ML | ML H ML M | VL M L ML | H A VL L | ML VH L ML | ML MH ML M | H A VL L | ML MH ML M |
| C4 | ML MH ML M | ML VH L ML | MH VH VL L | VL M L ML | ML H ML M | H A VL L | H A VL L | ML VH L ML | MH VH VL L |
| C5 | ML H ML M | MH VH VL L | ML VH L ML | ML VH L ML | MH VH VL L | ML H ML M | MH VH VL L | ML VH L ML | H A VL L |
| C6 | ML VH L ML | ML H ML M | MH VH VL L | H A VL L | ML H ML M | ML VH L ML | VL M L ML | MH VH VL L | ML MH ML M |
| C7 | VL M L ML | MH VH VL L | ML VH L ML | ML H ML M | H A VL L | H A VL L | H A VL L | VL M L ML | VL M L ML |
| C8 | H A VL L | H A VL L | H A VL L | MH VH VL L | VL M L ML | ML H ML M | H A VL L | ML MH ML M | ML MH ML M |
| C9 | ML MH ML M | ML VH L ML | ML H ML M | MH VH VL L | H A VL L | MH VH VL L | ML VH L ML | VL M L ML | H A VL L |

| | Alternative 7 (Turkey) | | |
|---|---|---|---|
| | DM1 | DM2 | DM3 |
| C1 | MH VH VL L | ML VH L ML | VL M L ML |
| C2 | VL M L ML | H A VL L | ML MH ML M |
| C3 | ML VH L ML | ML H ML M | H A VL L |
| C4 | H A VL L | MH VH VL L | ML H ML M |
| C5 | ML H ML M | ML MH ML M | ML VH L ML |
| C6 | MH VH VL L | MH VH VL L | H A VL L |
| C7 | VL M L ML | VL M L ML | ML H ML M |
| C8 | H A VL L | H A VL L | MH VH VL L |
| C9 | ML MH ML M | ML H ML M | ML H ML M |

**Table 8.** Direct relation IVIF matrix.

|  | D1 | D2 | D3 |
|---|---|---|---|
| D1 |  | ((0.33,0.53),(0.17,0.37)) | ((0.13,0.43),(0.20,0.30)) |
| D2 | ((0.37,0.57),(0.23,0.37)) |  | ((0.37,0.57),(0.23,0.37)) |
| D3 | ((0.47,0.60),(0.17,0.30)) | ((0.37,0.57),(0.23,0.37)) |  |

**Table 9.** Normalized relation IVIF matrix.

|  | D1 | D2 | D3 |
|---|---|---|---|
| D1 |  | ((0.29,0.46),(0.14,0.31)) | ((0.11,0.37),(0.17,0.26)) |
| D2 | ((0.31,0.49),(0.20,0.31)) |  | ((0.31,0.49),(0.20,0.31)) |
| D3 | ((0.40,0.51),(0.14,0.26)) | ((0.31,0.49),(0.20,0.31)) |  |

**Table 10.** Total relation IVIF matrix.

|  | D1 | D2 | D3 |
|---|---|---|---|
| D1 | ((0.25,4.12),(0.07,0.31)) | ((0.45,4.27),(0.20,0.58)) | ((0.28,3.98),(0.22,0.52)) |
| D2 | ((0.61,4.93),(0.26,0.58)) | ((0.33,4.42),(0.09,0.36)) | ((0.49,4.46),(0.26,0.58)) |
| D3 | ((0.69,5.03),(0.20,0.52)) | ((0.60,4.83),(0.25,0.58)) | ((0.27,4.21),(0.08,0.31)) |

**Table 11.** Weights of the items.

| Dimensions | Weights | Criteria | Local Weights | Global Weights |
|---|---|---|---|---|
| Human capital (dimension 1) | 0.327 | Advanced education (Criterion 1) | 0.325 | 0.106 |
|  |  | Environmental experience (Criterion 2) | 0.336 | 0.110 |
|  |  | Multiple directorships experience (Criterion 3) | 0.339 | 0.111 |
| Social capital (dimension 2) | 0.342 | Ties to political organizations (Criterion 4) | 0.324 | 0.111 |
|  |  | Ties to universities (Criterion 5) | 0.332 | 0.113 |
|  |  | Ties to other companies (Criterion 6) | 0.344 | 0.118 |
| Cognitive capabilities (dimension 3) | 0.331 | Sensing abilities (Criterion 7) | 0.329 | 0.109 |
|  |  | Seizing abilities (Criterion 8) | 0.328 | 0.109 |
|  |  | Reconfiguring abilities (Criterion 9) | 0.343 | 0.113 |

Table 15 presents the sensitivity analysis results. In order to test whether the weights in this study are determined correctly, nine different weights were used by changing the weights in themselves. In other words, analyses were undertaken with nine different weights. This table shows that the proposed method provides a coherent result for ranking the best alternative of corporate governance-based strategic analysis for the sustainable production in energy industry of emerging economies.

**Table 12.** IVIF decision matrix.

|  | A1 | A2 | A3 | A4 | A5 | A6 | A7 |
|---|---|---|---|---|---|---|---|
| C1 | ((0.43,0.70),(0.13,0.23)) | ((0.30,0.70),(0.20,0.30)) | ((0.40,0.63),(0.13,0.23)) | ((0.30,0.60),(0.17,0.27)) | ((0.37,0.70),(0.17,0.27)) | ((0.33,0.63),(0.17,0.27)) | ((0.30,0.60),(0.17,0.27)) |
| C2 | ((0.27,0.53),(0.17,0.27)) | ((0.23,0.53),(0.23,0.33)) | ((0.33,0.57),(0.20,0.30)) | ((0.33,0.57),(0.20,0.30)) | ((0.23,0.53),(0.23,0.33)) | ((0.23,0.50),(0.27,0.37)) | ((0.33,0.57),(0.20,0.30)) |
| C3 | ((0.50,0.73),(0.17,0.27)) | ((0.40,0.63),(0.13,0.23)) | ((0.40,0.63),(0.23,0.33)) | ((0.33,0.60),(0.20,0.30)) | ((0.33,0.63),(0.17,0.27)) | ((0.40,0.60),(0.23,0.33)) | ((0.40,0.70),(0.20,0.30)) |
| C4 | ((0.37,0.60),(0.23,0.33)) | ((0.50,0.70),(0.17,0.27)) | ((0.33,0.63),(0.17,0.27)) | ((0.37,0.63),(0.20,0.30)) | ((0.33,0.60),(0.20,0.30)) | ((0.47,0.73),(0.13,0.23)) | ((0.47,0.70),(0.17,0.27)) |
| C5 | ((0.37,0.67),(0.20,0.30)) | ((0.33,0.63),(0.17,0.27)) | ((0.23,0.53),(0.23,0.33)) | ((0.47,0.70),(0.17,0.27)) | ((0.37,0.67),(0.20,0.30)) | ((0.47,0.73),(0.13,0.23)) | ((0.30,0.60),(0.27,0.37)) |
| C6 | ((0.40,0.73),(0.17,0.27)) | ((0.40,0.60),(0.23,0.33)) | ((0.40,0.60),(0.23,0.33)) | ((0.30,0.63),(0.27,0.37)) | ((0.40,0.70),(0.20,0.30)) | ((0.30,0.53),(0.20,0.30)) | ((0.53,0.73),(0.10,0.20)) |
| C7 | ((0.40,0.63),(0.23,0.33)) | ((0.33,0.57),(0.20,0.30)) | ((0.33,0.57),(0.20,0.30)) | ((0.30,0.60),(0.17,0.27)) | ((0.50,0.73),(0.17,0.27)) | ((0.43,0.67),(0.13,0.23)) | ((0.17,0.47),(0.23,0.33)) |
| C8 | ((0.47,0.70),(0.17,0.27)) | ((0.33,0.57),(0.20,0.30)) | ((0.43,0.67),(0.13,0.23)) | ((0.60,0.80),(0.10,0.20)) | ((0.30,0.57),(0.20,0.30)) | ((0.40,0.60),(0.23,0.33)) | ((0.57,0.77),(0.10,0.20)) |
| C9 | ((0.37,0.63),(0.23,0.33)) | ((0.50,0.70),(0.17,0.27)) | ((0.47,0.67),(0.17,0.27)) | ((0.30,0.60),(0.27,0.37)) | ((0.53,0.73),(0.10,0.20)) | ((0.33,0.63),(0.17,0.27)) | ((0.30,0.67),(0.30,0.40)) |

**Table 13.** Defuzzified decision matrix.

|      | **A1** | **A2** | **A3** | **A4** | **A5** | **A6** | **A7** |
|------|--------|--------|--------|--------|--------|--------|--------|
| C1   | 0.75   | 0.75   | 0.70   | 0.67   | 0.75   | 0.70   | 0.67   |
| C2   | 0.62   | 0.67   | 0.70   | 0.70   | 0.67   | 0.68   | 0.70   |
| C3   | 0.83   | 0.70   | 0.80   | 0.72   | 0.70   | 0.78   | 0.80   |
| C4   | 0.77   | 0.82   | 0.70   | 0.75   | 0.72   | 0.78   | 0.80   |
| C5   | 0.77   | 0.70   | 0.67   | 0.80   | 0.77   | 0.78   | 0.77   |
| C6   | 0.78   | 0.78   | 0.78   | 0.78   | 0.80   | 0.67   | 0.78   |
| C7   | 0.80   | 0.70   | 0.70   | 0.67   | 0.83   | 0.73   | 0.60   |
| C8   | 0.80   | 0.70   | 0.73   | 0.85   | 0.68   | 0.78   | 0.82   |
| C9   | 0.78   | 0.82   | 0.78   | 0.77   | 0.78   | 0.70   | 0.78   |

**Table 14.** Ranking results.

|                | **$S_i$** | **$R_i$** | **$Q_i$** | **Ranking** |
|----------------|-----------|-----------|-----------|-------------|
| A1 (China)     | 0.281     | 0.110     | 0.163     | 1           |
| A2 (India)     | 0.415     | 0.111     | 0.506     | 4           |
| A3 (Brazil)    | 0.501     | 0.113     | 0.819     | 6           |
| A4 (Mexico)    | 0.408     | 0.106     | 0.288     | 3           |
| A5 (Russia)    | 0.419     | 0.111     | 0.517     | 5           |
| A6 (Indonesia) | 0.494     | 0.118     | 0.984     | 7           |
| A7 (Turkey)    | 0.356     | 0.109     | 0.284     | 2           |

**Table 15.** Sensitivity analysis results.

|                | **Case 1** | **Case 2** | **Case 3** | **Case 4** | **Case 5** | **Case 6** | **Case 7** | **Case 8** | **Case 9** |
|----------------|------------|------------|------------|------------|------------|------------|------------|------------|------------|
| A1 (China)     | 1          | 1          | 2          | 1          | 1          | 1          | 1          | 1          | 1          |
| A2 (India)     | 4          | 2          | 1          | 5          | 2          | 4          | 6          | 5          | 4          |
| A3 (Brazil)    | 6          | 6          | 6          | 4          | 6          | 7          | 4          | 7          | 7          |
| A4 (Mexico)    | 3          | 4          | 3          | 3          | 7          | 5          | 3          | 2          | 3          |
| A5 (Russia)    | 5          | 3          | 7          | 6          | 3          | 2          | 7          | 6          | 6          |
| A6 (Indonesia) | 7          | 7          | 5          | 7          | 5          | 6          | 5          | 4          | 5          |
| A7 (Turkey)    | 2          | 5          | 4          | 2          | 4          | 3          | 2          | 3          | 2          |

## 5. Discussions and Conclusions

Over the past few decades, energy consumption has increased in emerging economies, which can be relevant in terms of economy growth. However, the intense consumption of energy is impacted by the growth of population and the reliance on traditional energy sources, which put pressure on the long-term sustainability of these countries. The recognition of these issues has triggered a number of policy-level initiatives and actions of various stakeholders, changing objectives and strategies of energy companies towards the sustainability direction. Within this context, the need to combine corporate sustainability and corporate governance has attracted wider attention and triggered scientific debates. In spite of the increasing scientific discussion, the characteristics of governance and more specifically new board members, contributing to sustainable production in energy industry of emerging economies and assessment of these characteristics are not entirely clear. Thus, we set forth to define a set of criteria and dimensions for analyzing the corporate governance-based strategic approach to sustainability in the energy industry of emerging economies.

The proposed model integrates the IVIF sets, IVIF DEMATEL, and IVIF VIKOR methods. The model provides a more consistent assessment system, which allowed us to overcome current limitations of prevailing studies. Thus, the study contributes to the literature by extending investigations on corporate governance and sustainable production in energy industry. Specifically, a more comprehensive range of governance characteristics linked to sustainable energy production was considered. The adopted methodology reveals the most significant characteristics of boards of directors, contributing to sustainable production in energy industry and, thus, address the first research question. Moreover, to our best knowledge no prior application of MCDA methods in such specific area as presented in this study have been observed. The study involved highly experienced experts with managerial and functional expertise in global energy industry as well as emerging economies and, thus, let us answer the second research question.

The obtained results demonstrate the significance of social capital of board of directors. Moreover, the ties to other companies, reconfiguring capabilities, ties to universities, multiple directorships, and ties to political organizations are the most significant characteristics of the board of directors promoting sustainable energy production in energy industry. Previous studies have confirmed these results. While other studies have confirmed the value of social capital in energy industry [34], the discussions on the significance of such value and the interrelationships to the strategic objectives of the firms have remained. Thus, our study confirms that directors with ties to other companies shape the development directions of energy companies toward sustainable production. Through the available network with other firms, the directors gain knowledge and information related to sustainable production. Moreover, the sharing of knowledge inside the company contributes to the increased board capital and, subsequently, the value of the board. Finally, the directors gain experience related to the adoption of sustainable energy production in various situations. The reconfiguring abilities of decision makers, referring to the ability to combine and modify the company's resources and competencies [33] necessary for sustainable production, appears to be significant in the energy industry context. These abilities are significant in adapting to the changes of legal and social environment, shaping the energy industry. Considering the maturity of energy companies, our results echo other studies assuming that board members of mature organizations are better at reconfiguring [43]. The investigations conducted in China revealed that social ties of board members with universities are interrelated to the higher level of a firm's responsibility [41]. The universities have an important role in disseminating values, social norms, and culture in promoting environmental protection and sustainable production. Multiple directorships experience may significantly influence environmental policy and practice due to obtained knowledge, experience, and reputation. Thus, the affiliation of board members with other boards lead to lower environmental legal litigations and sustainable energy production. The investigations performed in emerging market contexts demonstrate the significance of political ties [41]. These ties appear significant in obtaining green subsidies and subsequently adoption of sustainable energy production.

Finally, the contribution of the study is seen in complementing the literature on energy industry in the context of emerging countries. The ranking results of energy industries of emerging economies were obtained considering board specific characteristics contributing to sustainable energy production and, thus, let us address the third research question. The results reveal that the high-capacity economies such as China and Turkey have greater sustainable production potential in terms of corporate governance. Apparently, external pressures and expectations put higher pressure on the energy companies of these countries to be engaged in sustainable production. This result could be generalized for energy industry of emerging economies.

The implication of this study is to support corporate governance of energy companies regarding how to improve sustainable production through characteristics of board members. Thus, the results are relevant for the decision makers responsible for the selection of new board members. Considering the roles of boards, the energy companies should seek board members with ties to other companies, reconfiguring capabilities, ties to universities, multiple directorships, and ties to political organizations. While boards of directors act as the catalyst for corporate governance in

protection of stakeholders' interests, these characteristics of board members are vital in shaping proactive sustainability strategy of energy companies. Moreover, the results are relevant for evaluation of corporate boards. The formal evaluation routine could consider these characteristics leading to the replacement or development decisions. Apparently, these actions would lead to stronger boards and address sustainability issues.

This study aimed to evaluate the corporate governance in energy companies. This significant topic plays a key role especially for emerging economies. The main reason is that these countries aim to develop their industries to provide economic growth. Therefore, in this process, they can take many significant risks to achieve this objective. Hence, corporate governance plays a key role especially for these economies. However, this study has some limitations. First, the investigation considered only E7 countries. Thus, future investigations could be extended in other emerging countries. In addition, the comparison with developed, G7, or MINT economies could be conducted. Second, the study considered only a limited number of criteria. Thus, the future studies could be extended by including other criteria (e.g., environmental experience, etc.). Third, the results of the study can be compared by considering other techniques.

**Author Contributions:** Conceptualization, W.Q. and S.Y.; data curation, R.K. and Z.H.; formal analysis, S.Y. and H.D.; funding acquisition, W.Q.; investigation, R.K., S.Y., H.D., and Z.H.; methodology, W.Q. and H.D.; resources, S.Y., H.D., and Z.H.; software, R.K.; visualization, R.K.; writing—original draft, W.Q., S.Y., H.D., and R.K.; Writing—review and editing, Z.H. and H.D. All authors have read and agreed to the published version of the manuscript.

**Funding:** This work was sponsored in part by Science and Technology Development Plan Project of Jilin Province of China (20190601025FG), National Social Science Foundation of China (19BJL057), Social Science Foundation Project of Jilin Province of China (2019c14). This work was sponsored in part by Zhejiang Social Science Fund (20NDQN305YB).

**Acknowledgments:** This work was sponsored in part by Science and Technology Development Plan Project of Jilin Province of China (20190601025FG), National Social Science Foundation of China (19BJL057), Social Science Foundation Project of Jilin Province of China (2019c14). This work was sponsored in part by Zhejiang Social Science Fund (20NDQN305YB).

**Conflicts of Interest:** The authors declare no conflict of interest.

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
