# Peer review of "Corporate Governance-Based Strategic Approach to Sustainability in Energy Industry of Emerging Economies with a Novel Interval-Valued Intuitionistic Fuzzy Hybrid Decision Making Model"

_sustainability, doi:10.3390/su12083307_

Round 1

Reviewer 1 Report

This manuscript examines the sustainability in the energy industry of emerging economies based on a corporate governance-based strategic approach. In this context, the authors apply an extension of a hybrid decision making model with interval-valued intuitionistic fuzzy sets is applied and define related criteria. The main conclusion is that the energy industry could provide more sustainable results than the conventional managerial policies by considering the social capital of board members. This is an interesting research topic in the field of sustainability in the energy industry in combination with decision-making analysis. It is also important for the readers of the journal.

Overall, I found this an interesting manuscript with useful data from the extended literature review considering the corporate governance-based strategic approach in relation to sustainable production in the energy industry. There are, however, elements that need to be clarified and further explained.

  1. A more in depth analysis and explanation of the theoretical model in the case study should be provided.
  2. A further analysis of the data is needed. In this framework, the selection of the criteria needs to be further justified.
  3. A more logical explanation regarding the application of the model in relation to the theoretical description is required (explanation of percentages presented in Table 2, the dimensions and other data of Tables 3-6, the local and the global weights of Table 11, etc.).
  1. A further interpretation of the results is needed.
  1. The sensitivity analysis should be further explained and discussed.
  2. Referring to the compared emerging economies, could they be regarded as economy alternatives?
  3. In the “Discussions and Conclusion” section, it seems that several issues are repeated from the theoretical “Theoretical background” section.
  4. The applicability of the results needs to be better highlighted in the conclusions.
  5. The uncertainties of the model should be incorporated into the analysis.

Additional comments and recommendations for the improvement of the manuscript:

Abstract

  1. Introduction

[Line 92] “energy production?”

  1. Theoretical background

[Line 156] “Agency theory suggests”

[Line 194] “The stakeholder’s theory suggests”

[Table 1] It should be located after the text where it is mentioned.

[Table 2] “Karabasevic  et al”. Please check all references within the text.

  1. Methodology

[Line 243] a, b, c, d need to be defined.

[Line 257] “formula” or “equation”? Please check the whole text.

[Line 266] “equations”

Equations 10, 11, 21: Please check the symbols

Equation 25: Qi?

Equation 27: A(M) needs to be explained

  1. Case study

[Line 340] “Tables 3-6”

Table 11: Please check the Local Weights for Dimension 3.

  1. Discussions and Conclusion.

General note: The original contribution of the research has to be demonstrated based on the analysis in relation to the research questions.

References

  • The references need to be checked.
  • The same format should be applied.
  • Each reference cited must be referenced in the text

Some examples:

[Lines 483 and 486] There are many references “Åberg et al 2019” in the text

[Line 509] “Chams N.” It is not the same in the text

[Line 511] “J. F. R. (2013)” ?.

[Line 517] “Dagdeviren, Metin 2010” In the text it is referenced Dagdeviren et al.

[Line 544] “Johannisson B., Huse M., Johannisson B., Huse M. 2000. Recruiting” The names are mentioned twice.

[Line 593] “Post, Corinne; Byron K. 2015” It is not mentioned in the text.

Author Response

Response Letter for Reviewer 1

Reviewer Comment 1: A more in depth analysis and explanation of the theoretical model in the case study should be provided.

Author Response 1: The study was improved according to the reviewer’s comments. For this purpose, the part of “2.1. Sustainability and Sustainable Production in Energy Industry” was developed by considering new paragraphs. Hence, it is thought that the theoretical model in the study was improved. Additionally, new studies are also cited in this section. This new part is given below.

“2.1. Sustainability and Sustainable Production in Energy Industry

Energy is very important in terms of social and economic development in the country. Thanks to energy, people can meet their daily needs such as warming. Therefore, it is accepted that there is a strong connection between energy and people's social happiness. In addition to the mentioned issue, energy plays an important role in the economic development of countries [14]. The main reason for this is that energy is a very important raw material of the industry. As can be seen here, there is a positive correlation between a country's energy consumption and the amount of production. Therefore, more energy is needed to increase the industrial production of countries. In this way, new investments will increase, and this will contribute to the economic growth of the country [15]. Another advantage of the mentioned issue is that it is possible to reduce the unemployment rate of the country by providing new employment opportunities. It is obvious that the energy industry plays a significant impact on sustainable development [16].

In this process, one of the most used energy types is non-renewable energies. In this context, these types of energy are formed as a result of burning fossil fuels. Oil, natural gas and nuclear are important examples of non-renewable energy sources. One of the most important reasons for preferring these types of energy is its low cost. On the other hand, easier storage of reserves is also an important source of choice for non-renewable energies. However, these types of energy also have a number of disadvantages. The extensive use of non-renewable natural resources significantly influences high emission levels, especially in industrialised countries and subsequently contribute to the warming climate [17]. Meanwhile, developing countries are facing increased levels of emission due to economies development and population grow. Sustainability in energy industry refers to energy sources which have minimal environmental impact while considering social dimension through reliable and affordable energy supplies [18].

In addition, renewable energy sources take their sources from nature like wind and sun. This situation has many advantages to the country. For example, since renewable energy sources do not release carbon into the atmosphere, they do not create environmental pollution. Thus, the sustainable energy production appears to be a solution decreasing the negative impact on human health and contributing to the vital ecological systems. The focus on sustainable energy production has switched the attention of both scholars and practitioners to the renewable energy due to a great amount of potential as well as economic considerations.  The sustainable energy production is seen as a way to respond to previous conflicts [19] and disasters such as the BP oil disaster in the Gulf of Mexico in 2010 [20]. Therefore, the adoption of sustainable energy production demonstrates social and environmental responsibilities [21].

On the firm level, energy companies are pressured to respond to sustainability issues such as environmental performance, social impact, resource efficiency and others [22]. It appears that the firms are facing a challenge to implement short-term and long-term objectives. First, the energy companies are expected to follow many guidelines, international initiatives and agreements (e.g. the Sustainable Development Goals) in order to gain trust of key stakeholders [18]. Second, the energy companies are expected to remain profitable, competitive and maintain the social licence to operate [21]. Moreover, the scholars observe that challenges for sustainable production in energy industry are linked to insufficient human resources, failure to involve the beneficiaries and excessive focus on tactical solutions. Given this context, the role of those entrusted with corporate governance appear to be significant in balancing conflicting goals. The scholars argue that the board of directors make a number of sustainability related decisions [23], however it is not entirely clear about their role for sustainable production in energy industry.”

Reviewer Comment 2: A further analysis of the data is needed. In this framework, the selection of the criteria needs to be further justified.

Author Response 2: Based on the reviewer’s comments, in the “2. Theoretical background” section, a new section is created with the name of “2.3. Literature on MCDM Methods”. The studies in which MCDM methods are used are evaluated in this part. In addition, the gap is the literature is underlined and the main contribution of this study to the literature is identified. These new parts are given below.

“Table 1 summarizes the similar studies in the literature. With respect to the human capital dimension, advanced education plays a key role to improve corporate governance [7]. Additionally, environmental experience can also contribute this situation [37]. Another important criterion in this framework is the multiple directorships experience [8]. On the other side, regarding social capital dimension, strong ties to the political organizations and universities can provide effective corporate governance [41]. Similar to this situation, the companies, which aim to improve corporate governance, should have effective ties to other companies [34]. Moreover, sensing and seizing abilities are also significant in this framework [43]. Finally, reconfiguring abilities in the company can be very helpful to improve the effectiveness in the corporate governance activities [33].

2.3. Literature on MCDM Methods

A considerable number of studies investigated limited number of board’s characteristics, what lets us conclude that more integrated approach in evaluation of new board members to transition toward the sustainable production in energy industry is needed. MCDA methods appear to be popular in solving complex issues in the high-skilled human resources field. However, the analysis of the literature revealed no prior application of MCDA methods in such specific area as presented in this study. For instance, Krishankumar et al. [13] focused on the personel selection problem for information technologies industry. In the analysis process, extended VIKOR under intuitionistic fuzzy set context and intuitionistic fuzzy AHP were considered. Karabasevic et al. [44] made also similar examination for the same industry with EDAS and SWARA.

In addition to these studies, Sang et al. [45] made an analysis for personnel selection problem. In this framework, knowledge-intensive enterprises were included in the scope of the analysis. The evaluation of these companies was carried out by Karnik–Mendel (KM) algorithm and fuzzy TOPSIS approaches. Parallel to this study, Kelemenis et al. [46] also tried to make evaluation regarding effective personnel selection. On the other side, Kabak [47] aimed to identify the significant points in the selection of personnel. This analysis has been performed for militaries. In this context, a fuzzy DEMATEL-ANP based multi criteria decision making approach has been proposed in the evaluation process of this study. Furthermore, Ji et al. [48] and Rouyendegh and Erkan [49] made also evaluation for different industries, such as manufacturing and education. TODIM and fuzzy ELECTRE methods are considered in these analyses. As a result of the literature evaluation, it is found that there is a strong relationship between the corportate governance and sustainable production. Hence, it is believed that sector-based evaluation can contribute the literature in this framwork. For this purpose, an evaluation is performed in this study for energy industry. Another important point is that there are limited studies in which MCDM methods are considered for the energy industry. Thus, by considering IVIF DEMATEL and IVIF VIKOR methods for energy industry in this study, it is aimed to improve originality.”

Reviewer Comment 3: A more logical explanation regarding the application of the model in relation to the theoretical description is required (explanation of percentages presented in Table 2, the dimensions and other data of Tables 3-6, the local and the global weights of Table 11, etc.).

Author Response 3: According to the reviewer’s comments, more logical explanation regarding the application of the model in relation to the theoretical description is given. For this purpose, the percentages presented in Table 2 are given. Additionally, the process in Tables 3-6 is explained. In the final stage, the meanings of the local and the global weights of Table 11 are given. These new parts are given below.

“Table 2 indicates the scores for linguistic evaluations. In the IVIF sets, belongingness and non-belongingness are taken into account in the analysis process. The total of them should be equation to 1. For this purpose, in Table 2, percentages are proposed for different linguistic terms for this framework. Decision makers give their linguistic priorities for each relation between criteria and dimensions. Tables 3-6 represent the evaluations for the dimensions and their criteria. Table 3 mainly gives information about the linguistic evaluations for the dimensions. For this purpose, 3 different experts made comparative evaluations to understand which dimensions are more significant. Similarly, Tables 4-6 explain these examinations for criteria under each 3 dimensions.”

“Similar procedures are also applied for the criteria of each dimension and then the weights of criteria and dimensions are calculated. The results are given in Table 11. In this table, both local and global weights are identified. The local weights explain the significance levels of the criteria in their own dimension. On the other side, global weights make a cumulative analysis and state these importance levels while considering whole criteria.”

“According to the results, social capital (dimension 2) has the highest importance among the dimensions, followed by cognitive capabilities (dimension 3) and human capital (dimension 1). However, the global weights of criteria illustrate that ties to other companies (Criterion 6) has the highest importance among criteria followed by reconfiguring capabilities (Criterion 9) and ties to universities (Criterion 6). In addition to them, multiple directorships (Criterion 3) and ties to political organizations (Criterion 4) are other significant criteria among the top five characteristics of the board of directors promoting sustainable energy production in energy industry. On the other side, it is also determined that advanced education (Criterion 1), sensing abilities (Criterion 7) and seizing abilities (Criterion 8) play a lower role in comparison with the other ones.”

Reviewer Comment 4: A further interpretation of the results is needed.

Author Response 4: Based on the reviewer’s comments, the results of the study are detailed. In this context, both IVIF DEMATEL and IVIF VIKOR results are extended.

Reviewer Comment 5: The sensitivity analysis should be further explained and discussed.

Author Response 5: Based on the reviewer’s comments, the sensitivity analysis is further explained. In this framework, the main reason and results of this sensitivity analysis are discussed. These new paragraphs are demonstrated below.

“The values of Qi are listed in ascending order to measure the rankings of energy industries of E7 countries considering board specific characteristics promoting sustainable energy production. Accordingly, the ranking results demonstrate that alternative 1 (China) has the best rank among the economies. Also, A7 (Turkey) and A4 (Mexico) are other successful countries in this purpose. Nevertheless, it is also concluded that alternative 6 (Indonesia) takes the worst place among the emerging economies. Similarly, A5 (Russia) and A3 (Brazil) are also less successful by comparing with others. However, sensitivity analysis is also applied for illustrating the coherencies of importance degrees for the criteria. Table 15 represents the ranking results of sensitivity analysis by 9 cases.”

“Table 15 indicates that sensitivity analysis. In order to test whether the weights in this study are determined correctly, 9 different weights were used by changing the weights in themselves. In other words, analyzes were made with 9 different weights. This table shows that proposed method provide a coherent result for ranking the best alternative of corporate governance-based strategic analysis for the sustainable production in energy industry of emerging economies.”

Reviewer Comment 6: Referring to the compared emerging economies, could they be regarded as economy alternatives?

Author Response 6: In the final paragraph of the discussion part, we explained again why wee used emerging economies in the analysis. After that, we gave necessary information about the limitations of this study. Finally, directions for the future studies are also underlined. In this context, we explained that our main limitation is focusing on only emerging countries. Hence, in the following studies, different countries can be taken into account. This new part is given below.

“This study aimed to evaluate the corporate governance in energy companies. This significant topic plays a key role especially for emerging economies. The main reason is that these countries aim to develop their industries to provide economic growth. Therefore, in this process, they can take many significant risks to achieve this objective. Hence, corporate governance plays a key role especially for these economies. However, this study has some limitations. First, the investigation considered only E7 countries. Thus, future investigations could be extended in other emerging countries. In addition, the comparison with developed, G7 or MINT economies could be conducted. Second, the study considered only limited number of criteria. Thus, the future studies could be extended by including other criteria (e.g. environmental experience and etc.). Third, the results of the study can be compared by considering other decision-making techniques. For instance, multi-criteria decision-making methods such as TOPSIS and MOORA can be applied in further researches.”

Reviewer Comment 7: In the “Discussions and Conclusion” section, it seems that several issues are repeated from the theoretical “Theoretical background” section.

Author Response 7: “Discussion and conclusion” section was modified. The repetition with the theoretical background eliminated. These new parts are demonstrated below.

“Over the past few decades, the energy consumption has increased in emerging economies, which can be relevant in terms of economy growth. However, the intense consumption of energy is impacted by the growth of population and the reliance on traditional energy sources, which put pressure on the long-term sustainability of these countries. The recognition of these issues has triggered a number of policy-level initiatives and actions of various stakeholders, changing objectives and strategies of energy companies towards sustainability direction. Within this context, the need to combine corporate sustainability and corporate governance has attracted the wider attention and triggered the scientific debates. In spite of the increasing scientific discussion, the characteristics of governance and more specifically new board members, contributing to sustainable production in energy industry of emerging economies and assessment of these characteristics are not entirely clear.  Thus, we set forth to define a set of criteria and dimensions for analyzing the corporate governance-based strategic approach to sustainability in the energy industry of emerging economies.

The proposed model integrates the interval-valued intuitionistic fuzzy sets, IVIF DEMATEL and IVIF VIKOR methods. The model provides more consistent assessment system, which let to us overcome current limitations of prevailing studies. Thus, the study contributes to the literature by extending investigations on corporate governance and sustainable production in energy industry. Specifically, more comprehensive range of governance characteristics linked to sustainable energy production is considered. The adopted methodology reveals the most significant characteristics of board of directors, contributing to sustainable production in energy industry and thus, address the first research question. Moreover, to our best knowledge no prior application of MCDA methods in such specific area as presented in this study have been observed. The study involved highly experienced experts with managerial and functional expertise in global energy industry as well as emerging economies and thus, let us answer the second research question.

The obtained results demonstrate the significance of social capital of board of directors. Moreover, the ties to other companies, reconfiguring capabilities, ties to universities, multiple directorships and ties to political organizations are the most significant characteristics of the board of directors promoting sustainable energy production in energy industry. Previous studies have confirmed these results. While other studies have confirmed the value of social capital in energy industry [34], the discussions on the significance of such value and the interrelationships to the strategic objectives of the firms have remained. Thus, our study confirms that directors with ties to other companies shape the development directions of energy companies toward sustainable production. Through the available network with other firms, the directors gain knowledge and information related to sustainable production. Moreover, the sharing of knowledge inside the company contributes to the increased board capital and subsequently, the value of board. Finally, the directors gain the experience related to the adoption of sustainable energy production in various situations. The reconfiguring abilities of decision-makers, referring to the ability to combine and modify the company’s resources and competencies [33, 60] necessary for sustainable production, appears to be significant in energy industry context. These abilities are significant in adapting to the changes of legal and social environment, shaping the energy industry. Considering the maturity of energy companies, our results echo other studies assuming that board members of mature organizations are better at reconfiguring [43]. The investigations conducted in China revealed that social ties of board members with universities are interrelated to the higher level of firms’ responsibility [41]. The universities take an important role in disseminating values, social norms and culture in promoting environmental protection and sustainable production. This type of social ties appears to be significant in emerging markets’ context where environmentalism is at an early stage. Multiple directorships experience may significantly influence environmental policy and practice due to obtained knowledge, experience and reputation. Thus, the affiliation of board members with other boards lead to lower environmental legal litigations [40] and sustainable energy production. The investigations performed in emerging market contexts demonstrate the significance of political ties [41]. These ties appear significant in obtaining green subsidies and subsequently adoption of sustainable energy production.

Finally, the contribution of the study is seen in complementing the literature on energy industry in emerging countries’ context. The ranking results of energy industries of emerging economies were obtained considering board specific characteristics contributing to sustainable energy production and thus, let us address the third research question. The results reveal that the high-capacity economies such as China and Turkey have greater sustainable production potential in terms of corporate governance. Apparently, external pressures and expectations put higher pressure on the energy companies of these countries to be engaged in sustainable production. This result could be generalized for energy industry of emerging economies. 

The implication of this study is to support corporate governance of energy companies how to improve sustainable production through characteristics of board members. Thus, the results are relevant for the decision makers responsible for the selection of new board members. Considering the roles of board, the energy companies should seek for board members with ties to other companies, reconfiguring capabilities, ties to universities, multiple directorships and ties to political organizations. While board of directors’ act as the catalyst for corporate governance in protection of stakeholders’ interests, these characteristics of board members are vital in shaping proactive sustainability strategy of energy companies.  Moreover, the results are relevant for evaluation of corporate board. The formal evaluation routine could consider these characteristics leading to the replacement or development decisions. Apparently, these actions would lead to the stronger board and address sustainability issues.”

Reviewer Comment 8: The applicability of the results needs to be better highlighted in the conclusions.

Author Response 8: The implications of the results were modified as follows:

The implication of this study is to support corporate governance of energy companies how to improve sustainable production through characteristics of board members. Thus, the results are relevant for the decision makers responsible for the selection of new board members. Considering the roles of board, the energy companies should seek for board members with ties to other companies, reconfiguring capabilities, ties to universities, multiple directorships and ties to political organizations. While board of directors’ act as the catalyst for corporate governance in protection of stakeholders’ interests, these characteristics of board members are vital in shaping proactive sustainability strategy of energy companies.  Moreover, the results are relevant for evaluation of corporate board. The formal evaluation routine could consider these characteristics leading to the replacement or development decisions. Apparently, these actions would lead to the stronger board and address sustainability issues.

Reviewer Comment 9: The uncertainties of the model should be incorporated into the analysis.

Author Response 9: The benefit of IVIF sets to minimize uncertainty is emphasized in both introduction and the analysis parts. In addition to this issue, the main advantages of DEMATEL and VIKOR approaches are also explained in these parts. These new sentences are given below.

“The main benefit of considering IVIF sets is minimizing uncertainties in the decision-making process. In the other hand, the relationship between the criteria can be evaluated owing to the DEMATEL methodology. Also, the advantage of VIKOR method is that more appropriate results can be reached because of considering maximum group utility and minimum individual regret.”

“In this study, a hybrid decision making approach based on interval-valued intuitionistic fuzzy sets is proposed to measure the weights of dimensions and criteria for the corporate governance and to rank the energy industry of emerging economies for the sustainable production. Accordingly, the IVIF DEMATEL is used for weighting the factors of the corporate governance and the IVIF VIKOR is applied for evaluating the emerging economy alternatives. The main advantage of using IVIF sets is that it can be possible to minimize uncertainty in the analysis process Additionally, owing to the DEMATEL approach, impact relation map can be generated. Moreover, compromise solutions can be achieved by considering the closeness to the ideal solutions.”

Reviewer Comment 10: Additional comments and recommendations for the improvement of the manuscript:

 Abstract

  1. Introduction

[Line 92] “energy production?”

  1. Theoretical background

[Line 156] “Agency theory suggests”

[Line 194] “The stakeholder’s theory suggests”

[Table 1] It should be located after the text where it is mentioned.

[Table 2] “Karabasevic  et al”. Please check all references within the text.

  1. Methodology

[Line 243] a, b, c, d need to be defined.

[Line 257] “formula” or “equation”? Please check the whole text.

[Line 266] “equations”

Equations 10, 11, 21: Please check the symbols

Equation 25: Qi?

Equation 27: A(M) needs to be explained.

  1. Case study

[Line 340] “Tables 3-6”

Table 11: Please check the Local Weights for Dimension 3.

  1. Discussions and Conclusion.

General note: The original contribution of the research has to be demonstrated based on the analysis in relation to the research questions.

References

  • The references need to be checked.
  • The same format should be applied.
  • Each reference cited must be referenced in the text

Some examples:

[Lines 483 and 486] There are many references “Åberg et al 2019” in the text

[Line 509] “Chams N.” It is not the same in the text

[Line 511] “J. F. R. (2013)” ?.

[Line 517] “Dagdeviren, Metin 2010” In the text it is referenced Dagdeviren et al.

[Line 544] “Johannisson B., Huse M., Johannisson B., Huse M. 2000. Recruiting” The names are mentioned twice.

[Line 593] “Post, Corinne; Byron K. 2015” It is not mentioned in the text.

Author Response 10: We have completed all these mistakes. Within this context;

  • We corrected all grammar mistakes.
  • We have explained all necessary terms
  • We have designed references according to the journal rule.
  • We have corrected all problematic symbols.
  • “Discussion and conclusion” section was rewritten.

Reviewer 2 Report

Title
Does the title accurately summarise the content?
The title “Corporate governance-based strategic approach to sustainability in energy industry of emerging economies with a novel interval-valued intuitionistic fuzzy hybrid decision making model”, summarises the content accurately.

Abstract
Does the Abstract adequately summarise the text?
The Abstract summarizes the text adequately because it begins with a brief introduction, mentions the main objective of the article, also mentions a brief description of the methodology and finally presents the main results.

Argument
Does the manuscript develop a logical argument?
Yes, the manuscript develops a logical argument. The authors present 5 sections: section 1 with the theoretical discussion on sustainability, sustainable production in energy industry and interrelationships of corporate governance and sustainable production; section 2, the methodological background; section 3 explains the methodology applied,
section 4 presents the case of study and results, and finally section 5 with a discussion and conclusion.

Is the argument readily apparent and clear to the reader?
Yes, the argument is readily apparent and clear to the reader, but the authors maybe can present a summary of the explanation of the methodology or maybe add schemes to explain it.

Line 357: Fix the page number.

Line 359: Add page number.

Table 11: It's cut in two parts, try to add the title in next page.

Are there sufficient case studies and examples to add colour, relevance and credibility?
Yes, the manuscript has sufficient case studies and examples to add colour, relevance and credibility.

Page 15: Add page number and fix all the page numbers.

Author Response

Response Letter for Reviewer 2

Reviewer Comments: Is the argument readily apparent and clear to the reader?

Yes, the argument is readily apparent and clear to the reader, but the authors maybe can present a summary of the explanation of the methodology or maybe add schemes to explain it.

Line 357: Fix the page number.

Line 359: Add page number.

Table 11: It's cut in two parts, try to add the title in next page.

Page 15: Add page number and fix all the page numbers.

Author Response: According to the reviewer’s comments, the following improvements have been performed.

On the line 357, the page number is fixed.

On the line 359, the appropriate page number is given.

Table 11 is demonstrated on the one page.

On the page 15, we added page number. Additionally, all page numbers are fixed in the order.

Round 2

Reviewer 1 Report

In the revised edition, the manuscript has been significantly improved and covered my main comments